# Self-supervised Learning for Speech Enhancement

**Yu-Che Wang** [* 1]  **Shrikant Venkataramani** [* 1]  **Paris Smaragdis** [1 2 3]

## Abstract

Supervised learning for single-channel speech enhancement requires carefully labeled training examples where the noisy mixture is input into the network and the network is trained to produce an output close to the ideal target. To relax the conditions on the training data, we consider the task of training speech enhancement networks in a self-supervised manner. We first use a limited training set of clean speech sounds and learn a latent representation by autoencoding on their magnitude spectrograms. We then autoencode on speech mixtures recorded in noisy environments and train the resulting autoencoder to share a latent representation with the clean examples. We show that using this training schema, we can now map noisy speech to its clean version using a network that is autonomously trainable without requiring labeled training examples or human intervention.

## 1. Introduction

Given a mixture of a speech signal co-occurring in a background of ambient noise, the goal of single-channel speech enhancement is to extract the speech signal in the given mixture. With recent advancements in Neural Networks (NNs) and deep learning, several neural network based approaches have been proposed for single-channel speech enhancement (Xu et al., 2013; Weninger et al., 2015; Pascual et al., 2017). These networks and approaches are predominantly trained in a supervised manner. The noisy mixture signal is fed as an input to the NN. The NN is then trained to estimate the corresponding clean speech signal in the mixture at its output. Thus, to train NNs for supervised speech enhancement, we require access to a vast training set of paired examples of noisy mixtures and their corresponding clean speech versions. As a result, supervised learning approaches

to speech enhancement and source separation suffer from the following drawbacks.

1. Clean targets can often be difficult or expensive to obtain. For example, bird calls recorded in a forest are often found to be in the presence of interfering sounds like the ones from animals, trees and thunderstorms. Alternatively, machine fault recordings are often taken when the machine is in operation to identify potential damages and it is infeasible to record these sounds in an isolated manner to obtain clean recorded versions.

2. These networks cannot be used as stand-alone learning machines that autonomously learn to denoise speech mixtures in ambient recording environments.

3. The trained speech enhancement systems can potentially be deployed in previously unseen conditions. Thus, there is a strong possibility of a mismatch between the training and test conditions. In such cases, we do not have the ability to use the recorded test mixtures to improve the performance of our model in the unseen test setting.

To relax the constraints of paired training data, a few recent approaches interpret the problem of denoising and source separation as a style-transfer problem wherein, the goal is to map from the domain of noisy mixtures to the domain of clean sounds (Stoller et al., 2018; Michelashvili et al., 2019; Venkataramani et al., 2019). These approaches only require a training set of mixtures and a training set of clean sounds, but the clean sounds can be unpaired and unrelated to the mixtures. However, these methods rely on training a pair of autoencoders jointly, one for each domain in order to learn the mapping and can be tedious to train. Other approaches have tried to relax the constraints by learning to enhance noisy mixtures in a "weakly supervised" setting. Instead of using representative clean training examples to identify a source, these approaches use alternate techniques for identification. For example, (Kong et al., 2020; Pishdadian et al., 2020) assume that in addition to the mixtures, we have access to information about when the source we wish to isolate is active in the mixture. Generating the timing information about the activity of the source requires training an event detection network which relies either on human listening or on clean training examples. Furthermore, these

---

[*]Equal contribution  [1]University of Illinois at Urbana-Champaign [2]Adobe Research [3]Supported by NSF grant #1453104. Correspondence to: Yu-Che Wang <yuchecw2@illinois.edu>.

*Proceedings of the 37th International Conference on Machine Learning*, Vienna, Austria, PMLR 119, 2020. Copyright 2020 by the author(s).

methods cannot be reused if the test conditions do not match the conditions under which the network was trained.

To relax constraints on training data, a recent learning paradigm gaining popularity in the fields of computer vision and natural language processing is the idea of self-supervised learning (Kolesnikov et al., 2019; Doersch & Zisserman, 2017; Lan et al., 2019). Instead of constructing large labeled datasets and using them for supervised learning, we use the relationships, correlations and similarities between the training examples to construct the corresponding paired labels for the training set. Thus, we can learn suitable representations and mappings from autonomously labeled training examples. In the case of audio, this strategy has been recently explored to learn unsupervised representations and perform speech recognition, speaker identification and other allied tasks (Pascual et al., 2019; Ravanelli & Bengio, 2019). However, self-supervision and unsupervised representation learning have not been explored for other audio applications including speech enhancement.

The goal of this paper is to develop and investigate the use of a self-supervised learning approach for speech denoising. To do so, we assume that we have access to a training set of clean speech examples. We first use these examples to learn a suitable representation for the clean sounds in an unsupervised manner. Thereafter, we use the learned representation along with noisy speech recordings to learn a mapping from the domain of mixtures to the domain of clean sounds. These developments allow us to devise speech enhancement systems that can learn autonomously in noisy ambient environments without human intervention, thereby alleviating the various drawbacks and constraints of supervised speech enhancement networks.

## 2. Self-Supervision for Speech Enhancement

As briefly discussed in Section 1, NN based supervised speech enhancement relies on the availability of paired training examples. This imposes several limitations on the trained networks and using self-supervision can relax these constraints. But first, we begin with a description of how we can train our NNs to perform speech enhancement in a self-supervised manner. To identify the source we wish to isolate from the mixtures, we assume that we have access to a dataset of clean sounds that represent the source. For example, if the goal is to isolate human speech from ambient noisy recordings, we assume that we have access to a dataset of a few clean speech examples. These examples can be completely unrelated to the mixture recordings used and contain a completely different set of speakers and utterances.

Over the last decade, a popular method to perform Self-supervised Speech Enhancement (SSE) problem is the idea

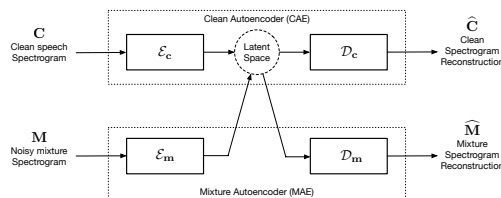

Figure 1. Block diagram of our self-supervised speech enhancement system. We first train the CAE to learn a latent representation for the clean sounds. We then autoencode on the mixtures and enforce that the MAE shares the latent space with the CAE using our cycle-consistency loss terms. Once both the autoencoders are trained, the diagonal path through $\mathcal{E}_{\mathbf{m}}$ and $\mathcal{D}_{\mathbf{c}}$ gives the denoised outputs at inference time.

of Non-negative Matrix Factorization (NMF). In the case of NMF based methods, the problem of SSE (also known as semi-supervised speech enhancement) was solved as a two-step procedure. In the first step, we perform an NMF decomposition of the clean sounds to learn representative spectral models for the speech signal. In the second step, we iteratively fit these models on unseen noisy speech recordings to isolate the underlying speech component from the ambient noise. However, NMF based SSE requires the use of an iterative fitting procedure for each test example during inference. We improve upon this by training NNs for SSE. To train NNs for SSE, we use a similar two-step approach.

1. In step I, we use the clean training examples to learn an unsupervised representation for the clean speech sounds. Essentially, we train an autoencoder NN on the magnitude spectrograms of the clean sounds and learn a suitable representation. We refer to this autoencoder as the Clean AutoEncoder (CAE).

2. In step II, we use ambient mixture recordings to train an autoencoder NN on the mixture spectrograms. We refer to this autoencoder as the Mixture AutoEncoder (MAE). The representations learned by the CAE is then used to modify the cost-functions used to train the MAE network so as to learn a shared space between the CAE and MAE representations. This allows us to learn a mapping from the domain of mixtures to the domain of clean sounds without paired training examples.

### 2.1. Network Architecture

Having described the overall outline of our SSE approach, we now begin with a description of the finer details. Figure 1 shows the block diagram of the proposed SSE approach. The network basically consists of a pair of Variational AutoEncoders (VAEs) and is motivated by the architecture for unsupervised domain translation (Liu et al., 2017). Here, $\mathcal{E}_{\mathbf{c}}$ and

$\mathcal{D}_{\mathbf{c}}$ denote the encoder and decoder for the CAE respectively. The magnitude spectrogram of the clean speech signal is given as the input to the CAE and the CAE is trained to reconstruct the input magnitude spectrogram. Once we learn the unsupervised representation, we use ambient noisy mixture recordings and the CAE to train the MAE. $\mathcal{E}_{\mathbf{m}}$ and $\mathcal{D}_{\mathbf{m}}$ represent the encoder and decoder for the mixture autoencoder. The cost-functions described in Section 2.2 enforce that the MAE learns a latent representation that is shared with the latent representation of the CAE. Once the MAE is also trained, the path $\mathcal{E}_{\mathbf{m}} \rightarrow \mathcal{D}_{\mathbf{c}}$ gives the enhanced speech component corresponding to the mixture spectrogram $\mathbf{M}$.

## 2.2. Cost-function

We now describe the cost-functions used to train our network.

### 2.2.1. TRAINING THE CAE

As seen earlier, the first step of the SSE is to train the CAE and learn a suitable representation for the clean sounds. To achieve this, we train the CAE by minimizing an appropriate measure of discrepancy between the input spectrogram $\mathbf{C}$ and its reconstruction $\widehat{\mathbf{C}}$. Here, we use the $L2$ norm of the error given by $\mathcal{L}_{\text{CAE}} = \left\| \mathbf{C} - \widehat{\mathbf{C}} \right\|_2^2 + \lambda_1 \cdot \mathcal{L}_{\text{KL-CAE}}$. Being a VAE, the goal of $\mathcal{L}_{\text{KL-CAE}}$ is to learn a latent representation that is close to a zero-mean normal distribution.

### 2.2.2. TRAINING THE MAE

Once we train the CAE, we now use the ambient mixture recordings, the CAE and ambient noise recordings to train the MAE. Since the MAE encounters different types of input signals, the cost-functions used to train the MAE can be divided into the following terms.

**Reconstruction Loss:** Given the mixture spectrogram $\mathbf{M}$ of a speech signal in the background of ambient noise, we feed $\mathbf{M}$ as an input to the MAE. We train the MAE to reconstruct the mixture spectrogram at its output and produce a reconstruction $\widehat{\mathbf{M}}$. As before, we use the $L2$ norm of the error given by $\mathcal{L}_{\mathbf{M}} = \left\| \mathbf{M} - \widehat{\mathbf{M}} \right\|_2^2$ as our cost-function.

**Cycle Loss:** We now describe the cost-function terms used to enforce a shared latent representation between the MAE and the CAE. To achieve this, we use the CAE and incorporate the following cycle-consistency terms into our cost-function. Given a mixture spectrogram $\mathbf{M}$, let $\mathbf{h}_M$ denote the corresponding latent representation at the output of the MAE encoder $\mathcal{E}_{\mathbf{m}}$. We can pass the latent representation $\mathbf{h}_M$ through the CAE decoder $\mathcal{D}_{\mathbf{c}}$ to get the clean version of the mixture spectrogram $\mathbf{C}_M$. This resulting spectrogram can be mapped back into the latent space through the

CAE encoder $\mathcal{E}_{\mathbf{c}}$ to get the latent representation $\widehat{\mathbf{h}}_M$. This can again be passed through the MAE decoder $\mathcal{D}_{\mathbf{m}}$ to get the reconstruction $\widehat{\mathbf{M}}$. Summarizing these in the form of equations, we now have,

$$\mathbf{h}_M = \mathcal{E}_{\mathbf{m}}(\mathbf{M}) \qquad \mathbf{C}_M = \mathcal{D}_{\mathbf{c}}(\mathbf{h}_M)$$
$$\widehat{\mathbf{h}}_M = \mathcal{E}_{\mathbf{c}}(\mathbf{C}_M) \qquad \widehat{\mathbf{M}} = \mathcal{D}_{\mathbf{m}}(\widehat{\mathbf{h}}_M)$$

With these relationships, we now enforce that the cycle reconstruction of the mixture spectrogram $\widehat{\mathbf{M}}$ resembles the input mixture spectrogram $\mathbf{M}$. Likewise, we also enforce that the two latent representations before and after the cycle loop through the CAE are close. Thus, the overall cycle loss term can be given as,

$$\mathcal{L}_{\text{cyc}} = \left\| \mathbf{M} - \widehat{\mathbf{M}} \right\|_2^2 + \lambda_2 \cdot \left\| \mathbf{h}_M - \widehat{\mathbf{h}}_M \right\|_2^2 \qquad (1)$$

**Noise Example Loss:** As we discuss in Section 2.3, one of the advantages of SSE is its ability to autonomously train in an ambient environment and learn to separate speech signals from their noisy backgrounds. To do so, we assume that the model also sees glimpses of the background without any speech signal. Such clips can be easily separated from clips that contain a mixture of speech and background noise using a simple thresholding operation on the energy of the signals. Given a noise input spectrogram $\mathbf{M}_N$, $\mathbf{h}_N$ denotes the corresponding latent representation and $\mathbf{C}_N$ denotes the clean version of the noise spectrogram. The latent representation can be reconstructed through the MAE decoder to get $\widehat{\mathbf{M}}_N$. As before, we now have the following relationships,

$$\mathbf{h}_N = \mathcal{E}_{\mathbf{m}}(\mathbf{M}_N) \qquad \mathbf{C}_N = \mathcal{D}_{\mathbf{c}}(\mathbf{h}_N)$$
$$\widehat{\mathbf{M}}_N = \mathcal{D}_{\mathbf{m}}(\mathbf{h}_N)$$

We now enforce $\mathbf{M}_N$ and $\widehat{\mathbf{M}}_N$ to be identical and $\mathbf{C}_N$ reduces to silence. The overall noise example loss term becomes,

$$\mathcal{L}_{\text{N}} = \left\| \mathbf{M}_N - \widehat{\mathbf{M}}_N \right\|_2^2 + \lambda_3 \cdot \left\| \mathbf{C}_N - \mathbf{0} \right\|_2^2 \qquad (2)$$

**Overall MAE Cost-function:** The overall function cost-function used to train the MAE is now a combination of the above loss terms. The overall cost-function also includes a term $\mathcal{L}_{\text{KL-MAE}}$ to enforce that the latent representations close to a zero-mean normal distribution.

$$\mathcal{L}_{\text{MAE}} = \mathcal{L}_{\mathbf{M}} + \mathcal{L}_{\text{cyc}} + \mathcal{L}_{\text{N}} + \lambda_4 \cdot \mathcal{L}_{\text{KL-MAE}}$$

## 2.3. Advantages of Self-supervision

Having seen the network architecture and the cost-functions used, we can now begin to understand the advantages of our proposed SSE approach. We enumerate these advantages below:

1. To train our SSE network, we only need access to a small dataset of clean speech examples to train our CAE and ambient mixtures and noise recordings to train our MAE. Thus, we do not require any paired training data unlike supervised speech enhancement methods.

2. Once the CAE is trained, the model only relies on mixtures and noise recordings for further training. These recordings can be directly obtained from the place of deployment. Thus, we now have a way of using unseen test mixtures to improve separation performance. This is beneficial particularly when there is a mismatch between the training and deployment environments.

3. With this training strategy, we can train our SSE network without any human intervention autonomously to enhance speech signals.

4. Once we train the CAE, we do not need access to clean speech examples further. All future training is completely dependent on the pre-trained CAE. When deploying the model in a test location, we need not transport data to different deployment locations. This is particularly advantageous from a security standpoint.

5. An added advantage we gain is the reusability of the CAE. The pre-trained CAE can be reused to perform SSE in different speech environments irrespective of the nature of the interfering sounds as seen in our experiments described in Section 3.

## 3. Experiments

We now present the details of our two experiments to evaluate the performance our trained SSE model.

### 3.1. Experimental Setup

To perform SSE using our network, we operate on the magnitude spectrograms of the mixtures and clean sounds. To compute these magnitude spectrograms, we use a window and DFT size of 1024 samples at a hop of 256 samples with a Hann window. The resulting magnitude spectrograms have 513 frequency bins for each frame.

The CAE networks used for our experiments consist of a cascade of 1D convolutional layers each. The CAE encoder $\mathcal{E}_c$ consists of a sequence of 4 1D convolutional layers where the size of the hidden dimension sequentially decreases from $513 \rightarrow 512 \rightarrow 256 \rightarrow 128 \rightarrow 64$. The CAE decoder $\mathcal{D}_c$ also consists of a cascade of 4 transposed convolutional layers where the size of the latent dimensions increase in the reverse order. Thus, the latent space is chosen to have a dimensionality of 64. We use a stride of 1 sample and a kernel size of 7 for the convolutions. Each convolutional layer is followed by a batch-norm layer and a softplus non-linearity. In case of the encoder $\mathcal{E}_c$, the we also add an EQ

norm layer after the soft-plus non-linearity. The task of the EQ-norm layer is to compute the mean of all the frames of its input separately for each input spectrogram in the batch and subtract the same.

The architecture of our MAE network also follows a similar strategy. The MAE encoder $\mathcal{E}_m$ comprises 6 1D convolutional layers where the hidden layer sizes decrease from $513 \rightarrow 512 \rightarrow 400 \rightarrow 300 \rightarrow 200 \rightarrow 100 \rightarrow 64$. The MAE decoder aims to invert this operation and consists of 1D transposed convolutions that increase in hidden layer sizes in the reverse way. As before, we use a stride and kernel size of 1 and 7 respectively. Also, each convolutional layer is succeeded by a batch-norm and a softplus activation function. Similar to $\mathcal{E}_c$, the MAE encoder $\mathcal{E}_m$ also includes an EQ norm layer after the soft-plus non-linearity.

To evaluate the SSE model, we use Perceptual Evaluation of Speech Quality (PESQ) (Rix et al., 2001) and composite metrics that approximate the Mean Opinion Score (MOS) including CSIG: predictor of signal distortion, CBAK: predictor of background intrusiveness, and COVL: predictor of overall speech quality (Hu & Loizou, 2008).

### 3.2. Datasets

#### 3.2.1. EXPERIMENT 1: DAPS

The first experiment is aimed at evaluating the performance of our SSE model on real recordings taken in indoor ambient environments. For this experiment, we use the Device And Produced Speech (DAPS) dataset (Mysore, 2014). The dataset consists of real-world recordings of speech taken in environments like bedrooms, offices, conference rooms and living rooms which contribute to the overlapping ambient noise in the recordings. The dataset consists of 10 male and 10 female speakers each reading out 5 scripts. Each of these 100 recordings are available in a clean format and also in noisy environments. We divide the scripts into 3 disjoint segments: clean, mix and test. Similarly, the speakers are also divided into 3 disjoint segments: clean, mix and test. The scripts and speakers from the clean segments are used to train the CAE. The mix and test segments are used to train the MAE and evaluate the model respectively. Such a bifurcation leads to a completely different set of speech examples and speakers across the 3 segments. We choose these speakers and scripts randomly and ensure that the male and female speakers are distributed evenly across the segments.

#### 3.2.2. EXPERIMENT 2: BBC SOUND EFFECTS

The second experiment deals with evaluating our SSE model on ambient street noise available in the BBC Sound Effects dataset (BBC, 2015). For the speech signals, we use the signals from the DAPS dataset. We use the speech clips

| Environment | PESQ | | | | CSIG | | | | CBAK | | | | COVL | | | |
|---|---|---|---|---|---|---|---|---|---|---|---|---|---|---|---|---|
| | SS | 0% | 30% | 50% | SS | 0% | 30% | 50% | SS | 0% | 30% | 50% | SS | 0% | 30% | 50% |
| ipad_livingroom1 | 1.30 | 1.43 | 1.43 | **1.47** | 1.65 | **2.50** | 2.46 | 2.25 | 1.56 | 1.82 | 1.88 | **1.98** | 1.32 | **1.91** | 1.89 | 1.80 |
| ipad_bedroom1 | 1.37 | 1.49 | 1.51 | **1.52** | 1.56 | **2.53** | 2.31 | 2.34 | 1.56 | 1.89 | **1.98** | 1.96 | 1.30 | **1.96** | 1.86 | 1.88 |
| ipad_confroom1 | 1.37 | 1.52 | **1.59** | **1.59** | 1.62 | **2.67** | 2.32 | 2.28 | 1.66 | 1.93 | 2.01 | **2.06** | 1.35 | **2.04** | 1.91 | 1.89 |
| ipad_office1 | 1.22 | 1.37 | **1.39** | 1.37 | 1.46 | **2.32** | 2.15 | 1.84 | 1.40 | 1.83 | **1.86** | 1.85 | 1.17 | **1.78** | 1.71 | 1.53 |
| ipad_office2 | 1.33 | 1.37 | 1.33 | **1.42** | 1.52 | **2.46** | 2.23 | 2.39 | 1.44 | 1.76 | 1.71 | **1.91** | 1.25 | 1.84 | 1.71 | **1.85** |
| ipadflat_confroom1 | 1.45 | 1.38 | 1.47 | **1.54** | 1.36 | 2.20 | **2.33** | 2.25 | 1.64 | 1.74 | 1.93 | **2.00** | 1.22 | 1.71 | 1.84 | **1.85** |
| ipadflat_office1 | 1.26 | 1.35 | 1.36 | **1.40** | 1.15 | **2.46** | 2.10 | 1.82 | 1.42 | 1.85 | 1.84 | **1.91** | 1.06 | **1.85** | 1.67 | 1.54 |
| iphone_livingroom1 | 1.38 | 1.30 | **1.42** | 1.40 | 1.24 | 2.11 | **2.27** | 2.09 | 1.57 | 1.78 | 1.85 | **1.90** | 1.14 | 1.64 | **1.79** | 1.69 |
| iphone_bedroom1 | 1.43 | 1.33 | 1.43 | **1.47** | 1.13 | **2.14** | 2.13 | 1.88 | 1.58 | 1.79 | 1.91 | **1.93** | 1.08 | 1.68 | **1.73** | 1.62 |

*Table 1.* DAPS experiment results. We compare the results of our SSE model with those of spectral subtraction (SS). We consider three versions of our our SSE model based on the amount of pure noise examples seen by the model during training viz., 0%, 30% and 50% as a percentage of the training data. Higher scores are better for all metrics. We see that our SSE models consistently outperform SS on all the metrics. In addition, increasing the noise percentage also improves upon the quality of the extracted speech signal and suppression of the interfering noises.

| City | SNR (dB) | PESQ | | | | CSIG | | | | CBAK | | | | COVL | | | |
|---|---|---|---|---|---|---|---|---|---|---|---|---|---|---|---|---|---|
| | | Mixture | 0% | 30% | 50% | Mixture | 0% | 30% | 50% | Mixture | 0% | 30% | 50% | Mixture | 0% | 30% | 50% |
| London | 5 | 1.09 | 1.32 | 1.31 | **1.36** | 1.96 | 2.02 | **2.03** | 1.97 | 1.69 | 1.99 | 2.06 | **2.13** | 1.43 | 1.58 | **1.61** | 1.58 |
| | 10 | 1.18 | 1.52 | 1.59 | **1.60** | 2.41 | 2.44 | **2.49** | 2.48 | 1.99 | 2.26 | 2.41 | **2.48** | 1.73 | 1.92 | **1.98** | 1.94 |
| Paris | 5 | 1.09 | 1.21 | **1.23** | 1.22 | 1.77 | 1.83 | **1.92** | 1.87 | 1.69 | 1.90 | 1.97 | **1.98** | 1.29 | 1.42 | **1.46** | 1.43 |
| | 10 | 1.18 | 1.48 | 1.48 | **1.50** | 2.03 | 2.23 | 2.22 | **2.28** | 1.98 | 2.19 | 2.28 | **2.34** | 1.53 | 1.79 | **1.81** | 1.79 |

*Table 2.* BBC experiment results. Similar to the DAPS experiment, we compare the results of our SSE model at three different noise percentages 0%, 30% and 50%. Considering the significant presence of non-stationary sounds in street noise recordings, we do not use spectral subtraction as our baseline method. Instead we report the metric values for the mixtures for comparison. As before, increasing the percentage of pure noise examples enhances the noise suppression (as seen by the CBAK scores) and the quality of the extracted speech (PESQ).

from the clean segment to train the CAE and the speech clips from the mixture segment for the MAE. Mixture audios are composed by mixing the clean speech sounds with ambient noises from two cities (London and Paris) at 2 SNR settings (5 and 10dB) each. For each city, we choose 10 ambient noise files which add up to 45 minutes of noises approximately. The same noise files are used to produce mix and test segments. We emphasize that the network has never encountered mixtures of the test speakers or their utterances with the noise files used during training.

### 3.3. Results and Discussion

Table 1 presents the results of our experiments on the DAPS dataset. We use spectral subtraction (SS) as our baseline method and compare it with three versions of our SSE model (based on the percentage of pure noise recordings encountered during training). We observe a consistent improvement in performance over SS in all the metrics, and the model also improves as it comes across a higher percentage of pure noise sounds. The environments livingroom1 and office1 are relatively more reverberant compared to the other environments. Via informal listening tests, we observed that the final results are dereverberated as well. Thus, we can potentially use this training strategy for other allied tasks

like dereverberation or bandwidth extension.

Table 2 presents the results of our SSE experiments on the BBC dataset. Since the BBC noise recordings include non-stationary sounds from the streets, we compare SSE models with the mixture metrics. We also observe that the performance improvement is greater in the case of mixtures having a higher signal-to-noise ratios. As before, a higher noise percentage improves upon SSE performance further.

## 4. Conclusion

In this paper we developed and investigated the idea of self-supervision in a single-channel speech enhancement setup. To accomplish this, we first trained an autoencoder on clean speech signals and learned an appropriate latent representation. This latent representation was then used in a downstream speech enhancement task to train an autoencoder for noisy speech mixtures so that the two autoencoders shared their latent spaces. This allowed us to map the domain of noisy speech mixtures to the domain of clean sounds autonomously and without clean targets. Our experiments demonstrate the efficacy of our training approach in ambient indoor environments and in the presence of street noises.

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
