# OpenReview forum: "Self-supervised Learning for Speech Enhancement"
_ICML.cc/2020/Workshop/SAS — Submitted to SAS 2020_

### Official Review · AnonReviewer2 · 2020-06-28
**Interesting idea, but serious issues with evaluation and assumptions for "self-supervision"**

**Rating:** 3
**Confidence:** 5

**Review:**

# Introduction

The authors propose a self-supervised approach for speech enhancement, where instead of requiring paired clean and noisy speech examples, one should be able to train a separate model for clean speech and a model for mixtures. The latter seems to be affected by the amount of noise-only samples seen during training, but the relationship is not clear from the results.

Unfortunately, the paper has two main issues, which I highlight through the questions below.

# 1. Regarding non-stationary noise and the noise example loss:

"Such clips can be easily separated from clips that contain a mixture of speech and background noise using a simple thresholding operation on the energy of the signals."
You cannot always guarantee that this method will work, unless you have an idea of the SNRs of the mixtures in your training set. How would you make sure a noise example is not a mixture with very low SNR? Also, this method would not work with lots of types of non-stationary noise (imagine random construction noises and someone turns on a jackhammer, for an extreme example). The authors themselves mention non-stationary noise being an issue for the BBC dataset, but do not address this issue, which makes me assume the model was trained on *supervised* noise samples instead of using a simple algorithm as mentioned.

A related question: what would be the effect of selecting an innapropriate threshold for the noise example selection operation?

# 2. Regarding evaluation:

Why are you only comparing against spectral subtraction? It would have been interesting to see how the proposed method fares against supervised and semi-supervised models. Also, spectral subtraction is an entire set of methods, and not a single method, so a reference and explanation of which algorithm was used is important, especially since for most of the simpler spectral subtraction methods, the method used for detecting speech presence is the most important bit, with the simplest methods being based on simple energy-based thresholding.

This is relevant especially because the objective scores for the metrics are all very low, and it would have been interesting to contrast it to methods that have supervision. What are the PESQ scores for the noisy signals in the DAPS case? They must not be much lower than what is shown for all enhanced scores. On the BBC examples, where the mixture scores are reported, the improvements in many cases are very low and might not even be statistically significant.

Another very serious issue with evaluation is that the authors used the same noise files for creating the mix and test files for the BBC dataset, meaning the model had seen all those noises previously.

# Conclusion

While the idea presented by the authors is very interesting and tries to solve a very relevant problem in speech enhancement, the evaluation performed by the authors is not sufficient and has serious problems (such as using previously seen noises to evaluate the model). Additionally, the authors assumption that a simple energy thresholding method would be sufficient for selecting noise-only samples is wrong and does not seem to be used by the authors themselves in the paper.

---

### Official Review · AnonReviewer3 · 2020-06-29
**The idea is interesting. Experimentation could be improved perhaps with larger sets.**

**Rating:** 6
**Confidence:** 4

**Review:**

The idea of using self-supervised learning for speech enhancement task sounds good. Training the CAE network offline and adapting the MAE network with noise samples from deployment environment is a plausible use-case. The paper is well-written and enjoyable to read. The experiments section and baseline are weak in my view though.

It would have been great if the authors provided pointers to some audio samples before and after enhancement. The PESQ scores seem quite low and is difficult to judge the quality of enhancement by just looking at those numbers.
Section 3.2.2 - It is a bit disappointing to see same noise files used for adapting to a new environment and also for testing. Would have been good to see the results on held-out noise set.

Authors have not described the inference pipeline and left the readers to guess. Is it a cyclic process using CAE and MAE or straight through MAE? One would assume the results to be better with a combination of encoder part of MAE and decoder part of CAE.

Section 2.2
Reconstruction loss term also appears in the cycle loss. Do we need a separate reconstruction loss in MAE cost? Is there experimental evidence to suggest otherwise.


Some minor corrections:

Figure 1 - size too small.

Table 1 caption : of our our SSE -> of our SSE

---

### Official Review · AnonReviewer1 · 2020-06-29
**Potentially good paper, but missing comparison with existing methods, weak experimentation**

**Rating:** 4
**Confidence:** 4

**Review:**

# Summary

This work proposes a method for unpaired speech enhancement. In the nutshell, two VAEs are used. The VAEs encode the clean signal and the noisy one respectively. Then the cycle consistency loss applied to ensure that the latent space is shared between the two VAEs.

In general, this paper combines several methods that were previously explored. This is completely valid. Creative combinations of existing methods deserve exploration and should be interesting to the community. Unfortunately, the paper is not being upfront about this. The paper is mostly easy to follow. The literature review is somewhat scarce. It is missing many relevant works on cycle consistency, unpaired SE, SE with VAEs and such. The experimental section looks very preliminary: the dataset is very small and the only baseline looks very weak.


# Novelty

Many of the building blocks have been previously proposed, but the combination of VAEs with the cycle consistency loss for unpaired SE seems to be novel.

The are some of the previous publications on the cycle consistency for SE:
- CycleGAN-based speech enhancement for the unpaired training data, Yuan & Bao 2019
- Cycle-Consistent Speech Enhancement, Meng et al., 2018

VAEs were also explored for speech enhancement:
- Bando et al., 2017
- Leglaive et al., 2019

As I stated before, it is absolutely valid to explore the combinations of existing methods. But such a paper should be more upfront about this and propose a better contrast and comparison with previous methods.

# Clarity

The paper is clearly written and easy to follow. The model is explained very well. The result section could benefit from more extensive discussion. It is a bit unclear what are the goals of which of two experiments and what is the exact baseline tested.

In the Intro, Step 2 is too vague even for the introduction.

# Correctness

All the statements seem to be correct in the paper.

# Experimentation

I am not familiar with the datasets used in this paper, but they look rather small. The baseline in the first experiment looks very weak and there is no baseline in the second experiment.

This section could be improved by adding the comparison to the existing methods of unpaired speech enhancement. It would be also beneficial to have a baseline with the paired SE as some sort of the "speed of light".

The paper paper would also benefit from discussion of why best PESQ and CBAK achieved with 50% noise examples and CSIG and COVL with 0%. Have the authors experimented with even higher percentage of the noise examples to get even higher metrics?

It would be great to hear some audio samples.

# Typos

There are several minor typos and grammatical mistakes. Ones I spotted:
- Section 2.2.1: "Being a VAE, the goal..."
- Section 2.2.2: "Once we train the CAE, we now..."
-  Section 2.3: "Having seen the network architecture and the cost-functions used, we can now begin to understand..."
- Section 3.1: "the we"
- Section 3.1" "subtract the same"
-
# Conclusion

There is a potential for a great paper here. Unfortunately, for now I recommend reject. The main reasons are: weak literature review, failure to compare to similar methods, weak experimentation.

---

### Decision · Program_Chairs · 2020-07-01

**Decision:**

Reject

**Comment:**

Dear author(s),

Thank you very much for your submission at the ICML2020@SaS workshop (https://icml-sas.gitlab.io/). Based on the scores assigned by the reviewers, we regret to inform you that the paper was rejected. We got 26 submissions and we were only able to accept 13 papers. We invite you anyway to consider the feedback of the reviewers and to follow our upcoming workshop on July 17.